# Dietary 25-Hydroxycholecalciferol Supplementation from Day 85 of Gestation to Farrowing Enhances Performance, Antioxidant Capacity, and Immunoglobulins of Sows and Newborn Piglets

**DOI:** 10.3390/ani14233378

**Published:** 2024-11-23

**Authors:** Shenfei Long, Shad Mahfuz, Xiangshu Piao

**Affiliations:** 1Beijing Jingwa Agricultural Science and Technology Innovation Center, Beijing 101205, China; 2State Key Laboratory of Animal Nutrition and Feeding, College of Animal Science and Technology, China Agricultural University, Beijing 100193, China

**Keywords:** antioxidant capacity, 25-hydroxycholecalciferol, immunity, reproductive performance, sows

## Abstract

25-hydroxycholecalciferol has been widely used to increase the calcium and phosphorus absorption in lactating sows, while there have been only a few studies focusing on its effect on performance and health status of gestating sows and their offsprings. In this study, the dietary 25-hydroxycholecalciferol supplementation from day 85 of gestation was shown to effectively enhance the performance, antioxidant capacity, and immunoglobulin levels in sows and newborn piglets, which demonstrated that 25-hydroxycholecalciferol was an efficient additive for improving the reproductive performance and immune-antioxidant status of gestating sows during production.

## 1. Introduction

In the late gestation of sows, the fetal bone develops and grows rapidly, and therefore sufficient maternal nutrition should be supplied to meet the needs of the fetus [1]. Maternal nutrition intervention during gestation in sows alters the organ structure, influences prenatal and neonatal growth, and weight gains in newborn pigs [2]. During the gestation and lactation period, the lack of vitamin D_3_ leads to calcium (Ca) and phosphorus (P) metabolism disorders, sow dystocia, increased frailty, and eventually affects production efficiency [3]. Therefore, it is very important to quickly adjust the Ca and P metabolism of sows. Usually, dietary vitamin D_3_ moves to the liver, converts to 25-hydroxycholecalciferol under the action of 25-hydroxylase, transports to the kidney, and then converts to 1,25-dihydroxyvitamin D, which can bind to its receptors to regulate Ca and P metabolism in the body of sows [4]. There are many other forms of vitamin D use in swine nutrition, due to a higher absorption rate and the omission of one hydroxylation, and 25-hydroxycholecalciferol as a vitamin D_3_ substitute is considered to be more efficient [5]. The maximum allowed levels of vitamin D and 25-hydroxycholecalciferol in feed for pigs are 50 μg/kg feed [6]. During late gestation and lactation, ordinary vitamin D_3_ cannot directly supply the needs of sows, while diets supplemented with 50 μg/kg 25-hydroxycholecalciferol can quickly regulate the production of calcium-binding protein, and improve the absorption capacity of Ca and P, which can promote the bone development and performance in sows and piglets [7,8].

Based on the previous studies carried out in our lab, we found dietary 25-hydroxycholecalciferol supplementation improved bone biochemical and biomechanical properties, including Ca and alkaline phosphatase (ALP) in broilers [9], and maternal supplementation with 50 μg/kg 25-hydroxycholecalciferol could also increase the reproductive performance, bone development, and milk fatty acid composition in lactating sows and suckling piglets [8,9]. Moreover, the maternal 25-hydroxycholecalciferol supplementation might also benefit the microbiota community via enhancing the relative abundance of *Lactobacillus* and *Alloprevotella* in the large intestine of suckling piglets [10]. In addition, 25-hydroxycholecalciferol might also potentially enhance the antioxidant capacities in mice [11,12], duck breeders [13], and suckling piglets via increasing serum superoxide dismutase (SOD) content during lactation [10]. However, there are still some studies showing no significant impact or only a slight impact of partial or total replacement of vitamin D_3_ by 25-hydroxycholecalciferol [5].

To the best of our knowledge, little is known about the efficiency of maternal 25-hydroxycholecalciferol supplementation on performance, antioxidant capacity, immune function, and serum biochemical properties in gestating sows and their offspring. We hypothesis on the expected effects of dietary supplementation with 25-hydroxycholecalciferol on improving performance, antioxidant, and immune properties in gestating sows and their offspring. Therefore, the aim of the current study was to investigate the maternal 25-hydroxycholecalciferol supplementation from day 85 of gestation on performance, nutrient utilization, colostrum composition, immunoglobulin, antioxidant capacity, and serum biochemical properties in sows and newborn piglets.

## 2. Materials and Methods

The conduct of this experiment was at the Fengning Swine Research Unit of China Agricultural University (Chengdejiuyun Agricultural and Livestock Co., Ltd., Chengde, China) and the procedure was agreed by the China Agricultural University’s Institutional Animal Care and Use Committee (AW71012202-1-4, Beijing, China).

### 2.1. Experimental Design, Animals, Diets and Management

On day 85 of gestation, forty Landrace × Yorkshire gestating sows (average body weight of 241 ± 6.8 kg; average parity of 3.47 ± 0.6) were divided into two treatments with twenty replicates per treatment (according to back fat thickness, body weight, and parity). From day 85 of gestation to farrowing, sows were fed a normal vitamin D diet as control (containing 50 μg/kg vitamin D_3_; CON) (NRC, 2012 [14]), and a 25-hydroxycholecalciferol-supplemented diet (containing 50 μg/kg 25-hydroxycholecalciferol). The dosage was based on our previous study in our lab and the maximum allowed levels of vitamin D and 25-hydroxycholecalciferol in feed for pigs by the European Union [6]. Haineng Bioengineering Co., Ltd. (Rizhao, China) supplied the 25-hydroxycholecalciferol. The nutrient levels in the basal diet met the recommended requirements set by NRC (2012) [14] (Table 1). This diet was used through all the experimental period.

All the sows were housed individually in 2.1 × 0.6 m^2^ gestation stalls from day 85 to 107 of gestation. On day 107 of gestation, after bathing and disinfecting, sows were transferred to the farrowing room and housed individually in 2.0 × 3.0 m^2^ farrowing crates. The room temperature was maintained at 22.0 °C (± 1.0 °C), the light was supplied from 06:30 to 16:30 and heat lamps were supplied for newborn piglets. Before farrowing, sows were fed diets at 08:00 and 16:00 (2.0 kg/day) (50% of the total feed was provided at each feeding time). After farrowing, sows were fed diets ad libitum at 04:30, 10:30, and 16:30. The number of total born piglets, piglets born alive, and stillborn piglets, as well as the individual birth body weight (BW) of newborn piglets was recorded at birth according to Wu et al. [15]. The stillborn rate was calculated, as follows: stillborn rate (%) = number of piglets dead/numbers of total piglets born × 100% [16]. On day 85 of gestation, an electronic pound was used to measure the BW of sows and the ultrasonic device (Piglog105; SFK Technology A/S, Herlev, Denmark) was used to determine the backfat thickness (at the last rib head, 6 cm from the midline, P2) [17].

### 2.2. Experimental Samples Collection

Within 2 h post-farrowing, 4 reproductive glands (first and last on both sides, 15 mL of milk per gland) were used for the collection of colostrum samples and then stored at −20 °C immediately after collection.

From day 107 to 109 of gestation, the garb sampling technique and rectal palpation were used to collect fresh fecal samples of sows (about 2 kg feces/sow; n = 6) closed to the average parity and BW. The fecal samples were frozen at −20 °C immediately after collection until analysis. The feces of individual sow of 3 days were collected individually and dried at 65 °C for 72 h.

On the day of farrowing, the 10 mL heparinized vacutainer tubes (Becton Dickinson Vacutainer System, Franklin Lakes, NJ, USA) were used to collected blood samples (from the anterior vena cava) of these sows (near the average parity and BW; n = 6) and their offspring (newborn piglets; n = 6). These blood samples were centrifuged at 4 °C and 3000× *g* for 15 min to obtain the serum samples (stored at −20 °C) [18].

### 2.3. Chemical Analyses

The contents of protein, lactose, and fat in colostrum samples were measured via a Milkoscan System 4000 (Foss North America, Eden Prairie, MN, USA; AOAC, 1990) [18].

The feed and fecal samples from each treatment were collected and ground to pass through a 1 mm sieve before analysis. The composition of calcium and amino acids in feed samples, and the contents of dry matter (DM), crude protein (CP), ether extracts (EE), and ash in feed and fecal samples were measured using the methods described by AOAC (2012) [19]. The gross energy (GE) in feed and fecal samples was determined using an automatic isoperibolic oxygen bomb calorimeter (Parr 1281, Automatic Energy Analyzer; Moline, IL, USA). Organic matter (OM) was calculated using the following equation: OM = DM − ash. An atomic absorption spectrophotometer (Z-5000; Hitachi, Tokyo, Japan) was used to determine the content of chromium in feed and fecal samples. Neutral detergent fiber (NDF) was determined using a fiber analyzer (Ankom Technology, Macedon, NY, USA). The calculation used for apparent total tract digestibility (ATTD) of nutrient was conducted following Li et al. [20].

The immunoglobulin-specific kits (Sanwei Biological Engineering Co., Ltd., Weifang, China) were used to measure the contents of immunoglobulins (IgA, IgG, and IgM) in colostrum and serum of sows and piglets (n = 6). A spectrophotometer (Leng Guang SFZ1606017568, Shanghai, China) was used to determine the concentrations of total antioxidant capacity (T-AOC), catalase (CAT), glutathione peroxidase (GSH-Px), malondialdehyde (MDA), SOD, and tumor nuclear factor-α (TNF-α) in colostrum and serum by spectrophotometric methods according to the instructions of the kit’s manufacturer (Nanjing Jiancheng Bioengineering Institute, Nanjing, China). Commercial kits (Zhongsheng Beikong Biotechnology & Science Inc., Beijing, China) were used to test the Ca, tartrate-resistant acid phosphatase (TRAP), ALP, and insulin concentrations in serum and colostrum by colorimetry. A commercial ELISA test kit (Sino-UK institute of Biological Technology, Beijing, China) was used to measure the contents of osteocalcin (OC) and crosslap (CL) in serum and colostrum.

### 2.4. Statistical Analysis

The data were analyzed using the PROC MIXED of SAS (version 9.2; SAS Inst. Inc., Cary, NC, USA) [21] with sows or newborn piglets as the experimental unit. Using the univariate procedure of SAS to determine the normality and homogeneity of variance. The Student’s *t*-test was used to measure the parameters. The standard error of the mean was provided in all values. A trend for significance was defined at 0.05 ≤ *p* ≤ 0.10, while the significant difference was determined at *p* < 0.05.

## 3. Results

### 3.1. Reproductive Performance and Colostrum Composition

The supplementation of 25-hydroxycholecalciferol in the sow diets enhanced (*p* < 0.05) the average BW of newborn piglets (Table 2), and the colostrum fat and protein contents compared with those fed CON (Table 3). There were no significant differences between the two groups in the number of piglets alive, stillborn rate, litter weight, and colostrum lactose parameters.

### 3.2. The ATTD of Nutrients

According to Table 4, supplementation of 25-hydroxycholecalciferol in the sow diets enhanced (*p* < 0.05) the ATTD of CP, and tended to increase the ATTD of GE (*p* = 0.07), DM (*p* = 0.07), OM (*p* = 0.07), and EE (*p* = 0.06) compared with those fed CON. There was no significant difference between the two groups in ATTD of NDF.

### 3.3. Antioxidant Status

According to Table 5, supplementation of 25-hydroxycholecalciferol in the sow diets enhanced (*p* < 0.05) the concentration of SOD, and reduced (*p* < 0.05) the level of MDA in the serum of newborn piglets. Meanwhile, supplementation of 25-hydroxycholecalciferol in the sow diet also increased (*p* < 0.05) the content of SOD, and tended to increase (*p* = 0.06) the content of T-AOC and reduced (*p* = 0.09) the concentration of MDA in colostrum compared with those fed CON. There were no significant differences between the two groups in serum T-AOC, CAT, and GSH-Px levels in piglets, serum antioxidant parameters in sows, as well as the colostrum CAT and GSH-Px levels.

### 3.4. Immunoglobulin and TNF-α in Serum and Colostrum

According to Table 6, supplementation of 25-hydroxycholecalciferol in sow diets enhanced (*p* < 0.05) the concentrations of IgG and IgA in the serum of newborn piglets, and improved (*p* < 0.05) the concentration of IgG and decreased (*p* < 0.05) the concentration of TNF-α in the serum of sows compared with those fed CON. There were no significant differences between the two groups in serum IgM and TNF-α levels in piglets, the serum IgA and IgM parameters in sows, as well as colostrum immunoglobulins and TNF-α levels.

### 3.5. Biochemical Indicators in Serum and Colostrum

According to Table 7, supplementation of 25-hydroxycholecalciferol in sow diets improved (*p* < 0.05) the concentration of Ca in the serum of newborn piglets, and enhanced (*p* < 0.05) the concentration of ALP in the serum and colostrum of sows compared with normal vitamin D_3_. There were no significant differences between the two groups in serum ALP level in piglets, or the serum and colostrum Ca level in sows.

According to Table 8, maternal 25-hydroxycholecalciferol supplementation also increased (*p* < 0.05) the contents of insulin and CL in the serum of sows, and tended to increase the contents of CL (*p* = 0.10) and OC (*p* = 0.09) in the colostrum of sows compared with those fed CON. There were no significant differences between the two groups in serum in insulin, TRAP, CL, and OC levels of piglets, the serum TRAP and OC levels in sows, and the colostrum insulin and TRAP levels.

## 4. Discussion

The birth BW of newborn piglets was important for the growth of the piglets. In the current study, dietary 25-hydroxycholecalciferol supplementation in sows during gestation increased the average birth BW of newborn piglets compared with CON, which was in agreement with the studies conducted by Weber et al. [22] and Zhou et al. [23], who reported that maternal 25-hydroxycholecalciferol supplementation increased the birth BW of newborn piglets compared with sows fed normal vitamin D_3_. Coffey et al. [24] reported that dietary 25-hydroxycholecalciferol supplementation in gilts during gestation also increased litter size. However, the current study did not show an increase in litter size, which might be due to differences in the amount of 25-hydroxycholecalciferol and the parity of sows. Maternal 25-hydroxycholecalciferol supplementation could also increase the 25-hydroxycholecalciferol level in milk and the plasma of piglets [25], which indicated that 25-hydroxycholecalciferol could transfer from the serum of sows to their offspring by milk. The 25-hydroxycholecalciferol in sows and piglets could quickly regulate the production of calcium-binding protein, improve the absorption capacity of Ca and P, and therefore promote fetal bone development and improve the birth weight gain in newborn piglets [7,8]. During gestation and farrowing, since ordinary vitamin D_3_ could not directly meet the needs of sows, the dietary inclusion of 50 μg/kg of 25-hydroxycholecalciferol could meet and exceed the needs of sows, thus allowing newborn piglets to access more nutrients for growth and development [26].

In the present study, we found that maternal 25-hydroxycholecalciferol supplementation increased fat and protein contents in colostrum, and immunoglobulins (IgG or IgA) in the serum of piglets or sows, which was in agreement with Zhou et al. [23], who reported that dietary 25-hydroxycholecalciferol supplementation in sows improved total solids in breast milk, milk protein, and lactose. These findings reflected the composition of the colostrum as well as the immune function of sows and newborn piglets were improved by 25-hydroxycholecalciferol supplementation. The improved colostrum composition might be due to 25-hydroxycholecalciferol being able to increase the nutrient digestibility in sows and modulate the fat content and the fatty acid composition in milk [9]. Moreover, the 25-hydroxycholecalciferol could improve survival and phagocytic ability of white blood cells [27], which indicated that 25-hydroxycholecalciferol could help improve immune function in animals. It has been reported that maternal 25-hydroxycholecalciferol supplementation could improve the IgG content in the serum of sows or milk compared with normal vitamin D_3_ [9,28]. However, there is a lack of studies explaining the mechanism of 25-hydroxycholecalciferol on modulating IgG levels in sows and piglets, which still need to be further studied.

For sows, the maternal nutrient digestibility is related to the nutrient supply for fetal growth, reproductive performance, and colostrum or milk quality. In the current study, we found that dietary 25-hydroxycholecalciferol supplementation in sows increased the ATTD of CP, and tended to increase the ATTD of OM, DM, GE, and EE compared with sows fed normal vitamin D_3_. Our findings were partly in agreement with a study performed by Zhang et al. [9], who reported that the ATTD of Ca was higher in 25-hydroxycholecalciferol fed sows compared with vitamin D_3_-fed sows. A previous study showed that pigs supplemented with 25-hydroxycholecalciferol had a higher ATTD of nitrogen and ash [29], which indicates that the ATTD of CP and OM was improved by 25-hydroxycholecalciferol. The improved ATTD of CP might also be due to 25-hydroxycholecalciferol molecule promoting the release of phytate-bound protein [30], and therefore preventing the formation of protein–phytate complexes within the gut, rendering the protein more susceptible to breakdown by pepsin in the stomach [31]. Enhancing the vitamin D_3_ status in broilers by supplementing 25-hydroxycholecalciferol has been shown to improve sternum structure, mineral accretion, and protein solubility [32], these findings also helped to explain the increased ATTD of CP. However, this finding and the underlying mechanisms still need further investigation.

For gestating sows, the increased fetal growth and mammary gland development could cause severe oxidative stress (including the oxidant injury of lipid, DNA, and protein in cells) in sows and their offspring during late gestation and farrowing [33]. The current study found increased contents of T-AOC, SOD, and reduced amounts of MDA in the serum or colostrum of sows and newborn piglets. This reflected an improved antioxidant status and alleviation of oxidative stress by 25-hydroxycholecalciferol supplementation. Lipid peroxidation was directly illustrated by changes in the serum or colostrum MDA concentration due to MDA being the final product of lipid peroxidation. This allowed MDA to be used as a biomarker for radical-induced damage and endogenous lipid peroxidation [34]. In the present study, 25-hydroxycholecalciferol could reduce the MDA content, indicating that 25-hydroxycholecalciferol was beneficial for limiting lipid peroxidation in sows and newborn piglets. The T-AOC was a non-enzyme system that serves to alleviate oxidative stress, SOD was one of the antioxidant enzymes that prevented lipid, protein, and DNA oxidation during oxidative stress. The current study was in accordance with the studies performed by Sardar et al. [11] and Garcion et al. [12], who reported 25-hydroxycholecalciferol increased antioxidant function via modulating SOD and glutathione levels. 25-hydroxycholecalciferol has been shown as effective in improving body antioxidant status via enhancing the enzymatic antioxidant system in ducks [35]. Therefore, one possible explanation for the current finding might be that 25-hydroxycholecalciferol could activate the enzymatic antioxidant (via improving GSH-Px and SOD levels) and non-enzymatic antioxidant (via enhancing T-AOC levels) systems in the serum of sows, which could protect the sows and piglets against oxidative stress [10].

The current study also showed increased contents of insulin in the serum of sows supplemented with 25-hydroxycholecalciferol, which reflected the increased efficiency of glucose utilization. The increased insulin content was beneficial to the survival rate and efficient growth in piglets. A previous study showed low circulating 25-hydroxyvitamin D concentrations are associated with defects in insulin action and insulin secretion in people with prediabetes [36]. However, its mechanism still needs to be further studied. Serum Ca and P were generally not sensitive enough to detect problems with mineral imbalances due to the efficient homeostatic mechanism of Ca and P in the body [37]. In the current study, we found dietary 25-hydroxycholecalciferol supplementation in sows increased the content of Ca in the serum of newborn piglets, and ALP and CL contents in serum or colostrum of sows. The OC content also tended to increase in the colostrum of sows compared with CON. The CL and OC were biomarkers for bone Ca and P development in newborn piglets, which was reflected by the increased Ca content in piglets. The increased Ca in the serum of piglets and the increased content of ALP in the serum and colostrum of sows reflected Ca and P utilization was improved by 25-hydroxycholecalciferol supplementation. A previous study showed diets supplemented with 25-hydroxycholecalciferol in broilers improved bone biochemical and biomechanical properties, including Ca and ALP [4]. The reason for the increased Ca content in the serum of piglets might be due to 25-hydroxycholecalciferol ability to supply vitamin D_3_ and maintain Ca homeostasis during gestation and lactation in sows [22]. This finding was beneficial for improving the bone status of sows and bone quality of newborn piglets [23,38]. And this finding might also potentially reduce the incidence of rickets and osteoarthropathy while promoting normal ossification in cartilage [39]. The positive effects of 25-hydroxycholecalciferol supplementation on maternal and neonatal bone health parameters might be due to the increased mRNA expression of calciotropic genes [9]. In sows, vitamin D_3_ might influence the concentrations of OC and Ca as well as the activities of total ALP in plasma [40], which indicated that 25-hydroxycholecalciferol might also increase OC, Ca, and ALP levels in serum. Moreover, supplementation of vitamin D_3_ to broilers enhanced the ability of the enterocytes in the small intestine in transporting P into the plasma compartment, resulting in an increase in P absorption and retention in the body [41].

## 5. Conclusions

In conclusion, dietary 25-hydroxycholecalciferol supplementation from day 85 of gestation increased the protein and fat content in colostrum and average birth body weight of newborn piglets, and enhanced the apparent total tract digestibility of crude protein in sows. In addition, diets supplemented with 25-hydroxycholecalciferol also improved the antioxidant capacity, immunoglobulins, and calcium- and phosphorus-related biochemical properties in sows and newborn piglets.

## Figures and Tables

**Table 1 animals-14-03378-t001:** Composition and chemical composition of basal diets (g/kg, as-fed basis).

Items	
Maize	596.6
Soybean meal	194.8
Wheat bran	150.0
Soy oil	28.3
Dicalcium phosphate	6.2
Limestone	13.6
Salt	3.0
Chromic oxide	2.5
0.5% mineral–vitamin premix ^1^	5.0
Nutrient composition ^2^	
Gross energy, MJ/kg	16.92
Digestible energy, MJ/kg	13.81
Crude protein	161.5
Calcium	7.2
Digestible phosphorus	2.7
Lysine	7.9
Methionine	2.6
Threonine	5.9
Tryptophan	1.8
Valine	6.4
Dry matter	904.3
Ether extracts	48.5
Organic matter	853.3
Neutral detergent fiber	178.1

^1^ Premix provided the following per kg of diet: 12,000 IU vitamin A; 2000 IU (equivalent to 50 μg/kg) vitamin D_3_ for the control treatment or 50 μg 25-hydroxycholecalciferol for the experimental treatment; 24 IU vitamin E; 2.0 mg vitamin K_3_; 6.0 mg riboflavin; 4.0 mg pyridoxine; 2.0 mg thiamine; 24 ng vitamin B_12_; 30 mg niacin; 20 mg pantothenic acid; 3.6 mg folic acid; 0.4 mg biotin; 0.4 mg choline chloride; 96 mg iron; 8.0 mg copper; 40 mg manganese; 0.56 mg iodine; 120 mg zinc; 0.4 mg selenium. ^2^ The digestible energy and digestible phosphorus were calculated values, the rest were analyzed values.

**Table 2 animals-14-03378-t002:** Effects of maternal 25-hydroxycholecalciferol supplementation from day 85 of gestation on reproduction performance in sows and newborn piglets.

Items	CON ^1^	25-OH-D3 ^1^	SEM	*p*-Value
Sows				
Average backfat thickness, mm	20.50	20.40	0.86	0.68
Average parity	3.33	3.60	0.40	0.68
Average body weight, kg	240.17	241.97	6.80	0.87
Piglets				
Number of piglets alive, heads	10.33	10.50	0.71	0.88
Stillborn rate, %	13.93	8.30	3.48	0.73
Litter weight, kg	14.41	16.51	0.97	0.19
Average birth body weight of piglets, kg	1.39	1.57	0.06	0.04

Note: SEM, standard error of the mean (n = 20). ^1^ CON: 50 μg vitamin D_3_/kg feed; 25-OH-D3: 50 μg 25-hydroxycholecalciferol/kg feed.

**Table 3 animals-14-03378-t003:** Effects of maternal 25-hydroxycholecalciferol supplementation from day 85 of gestation on colostrum composition (%).

Items	CON ^1^	25-OH-D3 ^1^	SEM	*p*-Value
Lactose	2.54	2.39	0.14	0.96
Fat	3.20	4.89	0.27	0.01
Protein	13.84	17.80	0.44	<0.01

Note: SEM, standard error of the mean (n = 6). ^1^ CON: 50 μg vitamin D_3_/kg feed; 25-OH-D3: 50 μg 25-hydroxycholecalciferol/kg feed.

**Table 4 animals-14-03378-t004:** Effects of maternal 25-hydroxycholecalciferol supplementation from day 85 of gestation on apparent total tract digestibility of nutrients.

Items ^2^	CON ^1^	25-OH-D3 ^1^	SEM	*p*-Value
GE	0.79	0.83	0.01	0.07
EE	0.59	0.66	0.01	0.06
CP	0.80	0.83	0.01	0.02
DM	0.78	0.82	0.01	0.07
NDF	0.80	0.83	0.01	0.21
OM	0.80	0.84	0.01	0.07

Note: SEM, standard error of the mean (n = 6). ^1^ CON: 50 μg vitamin D_3_/kg feed; 25-OH-D3: 50 μg 25-hydroxycholecalciferol/kg feed. ^2^ EE, ether extracts; CP, crude protein; DM, dry matter; NDF, neutral detergent fiber; OM, organic matter; GE, gross energy.

**Table 5 animals-14-03378-t005:** Effects of maternal 25-hydroxycholecalciferol supplementation from day 85 of gestation on antioxidant status in sows and newborn piglets.

Items	CON ^1^	25-OH-D3 ^1^	SEM	*p*-Value
Serum of piglets ^2^				
T-AOC, U/mL	6.68	8.19	0.96	0.33
SOD, U/mL	111.05	127.19	4.13	0.05
CAT, U/mL	7.75	9.32	0.7	0.19
GSH-Px, U/mL	391.26	432.81	22.05	0.25
MDA, nmol/mL	6.44	5.17	0.24	0.02
Serum of sows ^2^				
T-AOC, U/mL	6.94	7.01	0.34	0.88
SOD, U/mL	142.92	142.68	4.11	0.97
CAT, U/mL	8.01	8.84	0.54	0.31
GSH-Px, U/mL	754.54	796.64	16.75	0.12
MDA, nmol/mL	4.42	4.11	0.25	0.41
Colostrum ^2^				
T-AOC, U/mL	3.80	4.86	0.29	0.06
SOD, U/mL	20.03	40.90	3.37	0.01
CAT, U/mL	2.19	1.70	0.17	0.11
GSH-Px, U/mL	197.45	192.20	10.19	0.73
MDA, nmol/mL	7.58	5.64	0.61	0.09

Note: SEM, standard error of the mean (n = 6). ^1^ CON: 50 μg vitamin D_3_/kg feed; 25-OH-D3: 50 μg 25-hydroxycholecalciferol/kg feed. ^2^ T-AOC: total antioxidant capacity; SOD: superoxide dismutase; CAT: catalase; GSH-Px: glutathione peroxidase; MDA: malondialdehyde.

**Table 6 animals-14-03378-t006:** Effects of maternal 25-hydroxycholecalciferol supplementation from day 85 of gestation on contents of immunoglobulins and tumor nuclear factor-α in sows and newborn piglets.

Items	CON ^1^	25-OH-D3 ^1^	SEM	*p*-Value
Serum of piglets ^2^				
IgA, g/L	0.39	0.59	0.05	0.05
IgG, g/L	4.99	6.67	0.26	<0.01
IgM, g/L	0.35	0.52	0.09	0.23
TNF-α, pg/mL	51.19	48.86	2.98	0.61
Serum of sows ^2^				
IgA, g/L	0.82	0.72	0.07	0.34
IgG, g/L	8.04	10.72	0.56	0.01
IgM, g/L	0.64	0.72	0.03	0.16
TNF-α, pg/mL	123.99	95.46	5.87	0.01
Colostrum ^2^				
IgA, g/L	2.18	1.75	0.19	0.16
IgG, g/L	17.04	16.85	0.82	0.88
IgM, g/L	1.14	1.51	0.18	0.18
TNF-α, pg/mL	117.85	109.95	5.19	0.32

Note: SEM, standard error of the mean (n = 6). ^1^ CON: 50 μg vitamin D_3_/kg feed; 25-OH-D3: 50 μg 25-hydroxycholecalciferol/kg feed. ^2^ IgA: immunoglobulin A; IgG: Immunoglobulin G; IgM: immunoglobulin M; TNF-α: tumor nuclear factor-α.

**Table 7 animals-14-03378-t007:** Effects of maternal 25-hydroxycholecalciferol supplementation from day 85 of gestation on the contents of alkaline phosphatase and calcium in sows and piglets.

Items	CON ^1^	25-OH-D3 ^1^	SEM	*p*-Value
Serum of piglets				
Alkaline phosphatase, U/L	859.78	919.30	37.56	0.33
Calcium, mmol/L	3.93	4.68	0.08	<0.01
Serum of sows				
Alkaline phosphatase, U/L	63.60	75.59	3.10	0.03
Calcium, mmol/L	2.80	2.95	0.16	0.51
Colostrum				
Alkaline phosphatase, U/L	1248.01	1423.96	25.75	0.01
Calcium, mmol/L	7.92	8.07	0.28	0.69

Note: SEM, standard error of the mean (n = 6). ^1^ CON: 50 μg vitamin D_3_/kg feed; 25-OH-D3: 50 μg 25-hydroxycholecalciferol/kg feed.

**Table 8 animals-14-03378-t008:** Effects of maternal 25-hydroxycholecalciferol supplementation from day 85 of gestation on biochemical indicators in serum and colostrum of sows and piglets.

Items	CON ^1^	25-OH-D3 ^1^	SEM	*p*-Value
Serum of piglets				
Insulin, μIU/mL	5.97	6.67	0.30	0.18
TRAP ^2^, U/L	12.20	9.44	1.29	0.21
CL ^2^, ng/mL	67.94	75.81	2.94	0.13
OC ^2^, ng/mL	4.40	4.27	0.25	0.74
Serum of sows				
Insulin, μIU/mL	7.61	8.86	0.28	0.01
TRAP ^2^, U/L	10.97	11.00	0.39	0.96
CL ^2^, ng/mL	89.99	100.48	3.27	0.05
OC ^2^, ng/mL	5.16	5.09	0.15	0.74
Colostrum				
Insulin, μIU/mL	22.83	25.55	1.53	0.25
TRAP ^2^, U/L	11.84	10.53	1.28	0.49
CL ^2^, ng/mL	62.04	75.72	5.19	0.10
OC ^2^, ng/mL	4.16	5.05	0.33	0.09

Note: SEM, standard error of the mean (n = 6). ^1^ CON: 50 μg vitamin D_3_/kg feed; 25-OH-D3: 50 μg 25-hydroxycholecalciferol/kg feed. ^2^ TRAP: tartrate-resistant acid phosphatase; CL: crosslap; OC: osteocalcin.

## Data Availability

The original contributions generated for this study are included in the article, further inquiries can be directed to the corresponding author.

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
