# Peer review of "Dietary 25-Hydroxycholecalciferol Supplementation from Day 85 of Gestation to Farrowing Enhances Performance, Antioxidant Capacity, and Immunoglobulins of Sows and Newborn Piglets"

_animals, 2024, doi:10.3390/ani14233378_

Round 1
Reviewer 1 Report
Comments and Suggestions for Authors
The study reports the effect of supplementation of 25-OH-D3 compared with nonhydroxylated Vit D3 in pregnant sows on the performance, immunity, etc. in sows and piglets. The study results are interesting. Here are some points for consideration.
L13: performance
L22-23: Vit D should be in same unit.
L51-53: Please explain why 25-OH-D3 is needed compared with Vit D3 with respect to limitation of the conversion process. There should be better molecular explanation here.
L119-119: it seems only one day feces was collected. Is it standard for the digestibility trial for determining the digestibility?
L260-275: The explanation that " 25-OH-D3 molecule promoting the release of phytate bound protein" may increase protein digestibility, but I am unclear how 25-OH-D3 can release phytate bound protein? I am not sure if the digestibility data based on one day collection is reliable.
L277-296: Could authors explain about the biochemical pathways that lead to improved antioxidant status and immune response by 25-OH-D3?
L297: please explain how insulin level increased by 25-OH-D3.
Author Response
Dear Sir
Good day. Thank you very much for your kind consideration with our submitted article and offering us the further opportunity to submit the revised manuscript. Please find here the point to point expert reviewer’s and editor’s comments with necessary changes as per suggested with this attached file. We have revised our manuscript for language and grammar checked by a native English speaker working in our University, and highlight the change with red color. We do thanks to skilled reviewers, academic editors and editorial board members as well for their critical evaluation to make the manuscript more effective for review process in Animals Journal.
Many thanks.
Sincerely yours,
Prof. Dr. Xiang Shu Piao,
Beijing Jingwa Agricultural Science and Technology Innovation Center
College of Animal Science and Technology, China Agricultural University, Beijing 100193, China
Corresponding Author,
Email: piaoxsh@cau.edu.cn.
Reviewer 1:
Comment: The study reports the effect of supplementation of 25-OH-D3 compared with nonhydroxylated Vit D3 in pregnant sows on the performance, immunity, etc. in sows and piglets. The study results are interesting. Here are some points for consideration.
L13: performance
Response: Revised. Please refer to Line 14
Comment: L22-23: Vit D should be in same unit.
Response: Thanks for the suggestion. We have changed 2000 IU/kg into 300 ug/kg according to the calculation.
Comment: L51-53: Please explain why 25-OH-D3 is needed compared with Vit D3 with respect to limitation of the conversion process. There should be better molecular explanation here.
Response: Thanks for the suggestion, we have revised according to the advice. Please refer to Line 58-62. There are many other form of vitamin D using in the swine nutrition, due to a higher absorption rate and the omission of one hydroxylation, 25-hydroxycholecalciferol as vitamin D3 substitute is considered to be more efficient [5]. The maximum allowed levels of vitamin D and 25-hydroxycholecalciferol in feed for pigs are 50 μg/kg feed, respectively [6].
Comment: L119-119: it seems only one day feces was collected. Is it standard for the digestibility trial for determining the digestibility?
Response: Thanks for the suggestion, we have checked and changed this part into “From d 107-109 of gestation, the garb sampling technique was used to collect fresh fecal samples of sows (about 2 kg feces/sow; n = 6) closed to the average parity and BW.” We collected fresh fecal samples of each sow for 3 days Please refer to Line 138-142.
Comment: L260-275: The explanation that " 25-OH-D3 molecule promoting the release of phytate bound protein" may increase protein digestibility, but I am unclear how 25-OH-D3 can release phytate bound protein? I am not sure if the digestibility data based on one day collection is reliable.
Response: Thanks for the suggestion, we collected the feces for 3 days, using the grab sample technique and rectal palpation. We stated that the improved ATTD of CP might also be due to 25-hydroxycholecalciferol molecule promoting the release of phytate bound protein, however, this still need further study to prove. We added this statement in Line 325-326.
Comment: L277-296: Could authors explain about the biochemical pathways that lead to improved antioxidant status and immune response by 25-OH-D3?
Response: Thanks for the suggestion, please refer to Line 348-350. 25-hydroxycholecalciferol has been shown as effective in improving body antioxidant status via enhancing the enzymatic antioxidant system in ducks (Ren et al., 2018). Therefore, one possible explanation for the current finding might be the 25-hydroxycholecalciferol could activate the enzymatic antioxidant (via improving GSH-Px and SOD levels) and non-enzymatic antioxidant (via enhancing T-AOC level) system in serum of sows, which could protect the sows and piglets against oxidative stress.
The reference was “Ren, Z. Z., Q. F. Zeng, J. P. Wang, X. M. Ding, S. P. Bai, Z. W. Su, Y. Xuan, and K. Y. Zhang. 2018. Effects of maternal dietary canthaxanthin and 25-hydroxycholecalciferol supplementation on antioxidant status and calcium-phosphate metabolism of progeny ducks. Poult. Sci. 97:1361–1367.”
Comment: L297: please explain how insulin level increased by 25-OH-D3.
Response: Thanks for the suggestion, please refer to Line 357-361. The increased insulin content was beneficial to the survival rate and efficient growth in piglets. A previous study showed low circulating 25-hydroxyvitamin D concentrations are associated with defects in insulin action and insulin secretion in persons with prediabetes [36]. However, its mechanism still need to be further studied. The reference was “Fahim, A.; Christine, B.; David, F.; Michael, P.C.; Feras, M.H.; Gerald, M.R. 2015. Low circulating 25-hydroxyvitamin D concentrations are associated with defects in insulin action and insulin secretion in persons with prediabetes. J Nutr. 145:714-719. doi: 10.3945/jn.114.209171.”
Reviewer 2 Report
Comments and Suggestions for Authors
Comments and Suggestions for Authors
After reviewing the manuscript entitled “Dietary 25-hydroxycholecalciferol supplementation from d 85 of gestation to farrowing enhances performance, antioxidant capacity and immunoglobulins of sows and newborn piglets”, the following suggestions were made it:
Simple Summary
Lines 10,12 and 14: The Simple Summary should not contain abbreviations, please remove the abbreviations 25-OH-D3. These abbreviations should be replaced by the full names.
Line 13: Change “erformance” to “performance”.
Abstract
Line 20: Change “2” to “two”. 20 sows per treatment means there were 20 replicates?
Lines 17-38: The abstract is well written and organized, no additional changes are required in this section.
Keywords: antioxidant status; 25-hydroxycholecalciferol; newborn piglets. These words used as keywords are the same as those previously used in the title of the manuscript. Keywords should be different from those in the title (but related to the topic) to broaden the reach of academic search engines in case the manuscript is later published.
Introduction
Line 54: Before describing the positive effects of dietary supplementation with 25-OH-D3, the authors should mention which other forms of administration or products with vitamin D have been used. Likewise, the authors should justify with more solidity and scientific evidence (including cost-benefit aspects) why 25-OH-D3 should be evaluated instead of other available forms and products with vitamin D.
Lines 55-57: Please add more details about the doses of 25-OH-D3 that have been tested, as well as the experimental periods used. This will help the reader better understand the importance of 25-OH-D3 supplementation at correct doses and periods.
Lines 59-68: The authors should provide a more detailed description of the experimental conditions used in the studies cited in these lines. For a correct description, the authors should indicate the experimental doses and periods used in each of these studies, which will help to identify possible sources of heterogeneity of results between studies. Likewise, the authors should also add contrasting studies in which the results obtained are neutral (no effects) or negative. This is particularly important to identify appropriate ranges that could be tested in future studies.
Line 72: Considering the background previously reviewed in the introduction, the authors should add a clear hypothesis on the expected effects of dietary supplementation with 25-OH-D3.
Line 72: Please avoid using abbreviations in the description of the objective.
Material and methods
Line 88: The authors should add a scientific explanation justifying the dose of 50 ug/kg 25-OH-D3. Why did they not use other dose? Why didn't they perform a dose-response study to estimate the optimal dose instead of using a random dose?
Line 105: Please specify what proportion of the total feed was provided at each feeding time?
Lines 118-121: Authors must justify with biological arguments and scientific references why they consider it valid to evaluate digestibility using a single sample from a single day. Generally, scientifically valid digestibility data are estimated using samples from several consecutive days.
Lines 140-152: Are there scientific references that support the methods used in these evaluations?
Lines 154-160: Statistical analyses were performed and described correctly. However, the authors should add and describe the statistical models used.
Results
Lines 163-165: The description in Tables 2 and 3 is incomplete and needs to be improved. The authors should also describe the variables for which no statistically significant effects were detected.
Line 175: The abbreviation ATTD should be changed to the full name. This correction applies to all titles and subtitles.
Lines 176-225: The interpretation and description of the results tables is very well done. Just a minor correction in this section, the authors should also describe the variables for which no statistically significant effects were detected.
Discussion
Lines 226-323: The entire discussion section is very well-written and organized. Furthermore, the discussion contains the necessary depth to explain in detail the results obtained (although little information is available on the specific topic evaluated). Therefore, I have no additional corrections in this section.
Conclusions
Lines 325-327: The conclusion drawn by the authors is consistent with the title, objectives, and results of the manuscript. However, abbreviations should not be used in this section. Please correct.
Author Response
Dear Sir
Good day. Thank you very much for your kind consideration with our submitted article and offering us the further opportunity to submit the revised manuscript. Please find here the point to point expert reviewer’s and editor’s comments with necessary changes as per suggested with this attached file. We have revised our manuscript for language and grammar checked by a native English speaker working in our University, and highlight the change with red color. We do thanks to skilled reviewers, academic editors and editorial board members as well for their critical evaluation to make the manuscript more effective for review process in Animals Journal.
Many thanks.
Sincerely yours,
Prof. Dr. Xiang Shu Piao,
Beijing Jingwa Agricultural Science and Technology Innovation Center
College of Animal Science and Technology, China Agricultural University, Beijing 100193, China
Corresponding Author,
Email: piaoxsh@cau.edu.cn.
Reviewer 2:
Comment:
Comments and Suggestions for Authors
After reviewing the manuscript entitled “Dietary 25-hydroxycholecalciferol supplementation from d 85 of gestation to farrowing enhances performance, antioxidant capacity and immunoglobulins of sows and newborn piglets”, the following suggestions were made it:
Simple Summary
Lines 10,12 and 14: The Simple Summary should not contain abbreviations, please remove the abbreviations 25-OH-D3. These abbreviations should be replaced by the full names.
Response: Revised, we have corrected all the abbreviations 25-OH-D3 in this manuscript and use their full names.
Comment: Line 13: Change “erformance” to “performance”.
Response: Revised. Please refer to Line 14.
Comment:
Abstract
Line 20: Change “2” to “two”. 20 sows per treatment means there were 20 replicates?
Response: Revised. Please refer to Line 21, there were 20 replicates.
Comment: Lines 17-38: The abstract is well written and organized, no additional changes are required in this section.
Response: Thanks.
Comment: Keywords: antioxidant status; 25-hydroxycholecalciferol; newborn piglets. These words used as keywords are the same as those previously used in the title of the manuscript. Keywords should be different from those in the title (but related to the topic) to broaden the reach of academic search engines in case the manuscript is later published.
Response: Revised. We have changed into “antioxidant capacity; 25-hydroxycholecalciferol; immunity”. Please refer to Line 43-44.
Comment: Introduction
Line 54: Before describing the positive effects of dietary supplementation with 25-OH-D3, the authors should mention which other forms of administration or products with vitamin D have been used. Likewise, the authors should justify with more solidity and scientific evidence (including cost-benefit aspects) why 25-OH-D3 should be evaluated instead of other available forms and products with vitamin D.
Response: Thanks for the suggestion, we have added the following statement in the introduction. Please refer to Line 58-61 about previous reference about other forms of administration or products with vitamin D have been used. We added that “There are many other form of vitamin D using in the swine nutrition, due to a higher absorption rate and the omission of one hydroxylation, 25-hydroxycholecalciferol as vitamin D3 substitute is considered to be more efficient [5].” The reference was from “Lütke‐Dörhoff, M.; Schulz, J.; Westendarp, H.; Visscher, C.; Wilkens, M. R. 2022. Dietary supplementation of 25‐hydroxycholecalciferol as an alternative to cholecalciferol in swine diets: A review. J Anim Physiol Anim Nutri. 106(6):1288-1305.
Comment: Lines 55-57: Please add more details about the doses of 25-OH-D3 that have been tested, as well as the experimental periods used. This will help the reader better understand the importance of 25-OH-D3 supplementation at correct doses and periods.
Response: Thanks for the suggestion, we have revised and added more information about the doses of 25-OH-D3. Please refer to Line 61-67. So the statement changed into “During late gestation and lactation, ordinary vitamin D3 cannot directly supply the needs of sows, while dietary supplemented with 50 ug/kg 25-hydroxycholecalciferol can quickly regulate the production of calcium-binding protein”. And we also added that “The maximum allowed levels of vitamin D and 25‐OHD3 in feed for pigs are 50 μg/kg feed, respectively [6].
Comment: Lines 59-68: The authors should provide a more detailed description of the experimental conditions used in the studies cited in these lines. For a correct description, the authors should indicate the experimental doses and periods used in each of these studies, which will help to identify possible sources of heterogeneity of results between studies. Likewise, the authors should also add contrasting studies in which the results obtained are neutral (no effects) or negative. This is particularly important to identify appropriate ranges that could be tested in future studies.
Response: Thanks for the suggestion, we have revised and added more information about other studies showed no significant impact or only a slight impact of partial or total replacement of vitamin D3 by 25-hydroxycholecalciferol in a reference. Please refer to Line 79-81.
Comment: Line 72: Considering the background previously reviewed in the introduction, the authors should add a clear hypothesis on the expected effects of dietary supplementation with 25-OH-D3.
Response: Thanks for the suggestion, we have revised according to the advice. Please refer to Line 85-87. We have added that “We hypothesis on the expected effects of dietary supplementation with 25-hydroxycholecalciferol on improving performance, antioxidant and immune properties in gestating sows and their offspring.”
Comment: Line 72: Please avoid using abbreviations in the description of the objective.
Response: Revised. We have changed the abbreviations in the description of the objective, we use the whole name 25-hydroxycholecalciferol.
Comment:
Material and methods
Line 88: The authors should add a scientific explanation justifying the dose of 50 ug/kg 25-OH-D3. Why did they not use other dose? Why didn't they perform a dose-response study to estimate the optimal dose instead of using a random dose?
Response: Revised. The dosage is based on previous study in our lab and also the maximum allowed levels of vitamin D and 25‐OHD3 in feed for pigs are 50 μg/kg feed, respectively (European Union, 2019). Please refer to Line 105-107.
Comment: Line 105: Please specify what proportion of the total feed was provided at each feeding time?
Response: Revised. We have specified the proportion (50%) of the total feed was provided at each feeding time. Please refer to Line 124-125.
Comment: Lines 118-121: Authors must justify with biological arguments and scientific references why they consider it valid to evaluate digestibility using a single sample from a single day. Generally, scientifically valid digestibility data are estimated using samples from several consecutive days.
Response: Thanks for the suggestion, we have revised according to the advice. From d 107-109 of gestation, the garb sampling technique and rectal palpation were used to collect fresh fecal samples of sows (about 2 kg feces/sow; n = 6) closed to the average parity and BW from 3 consecutive days. The feces of individual sow of 3 days were collected individually and dried at 65 °C for 72 h. Please refer to Line 138-142.
Comment: Lines 140-152: Are there scientific references that support the methods used in these evaluations?
Response: Revised. We have added scientific references that support the methods used in these evaluations. About the ATTD of nutrients, we refer to Li et al (2019) [20]. And the antioxidant, immunoglobulins levels were measured by spectrophotometric methods method according to the the instructions of the kit’s manufacturer. The contents of protein, lactose, and fat in colostrum samples were measured via a Milkoscan System 4000 (Foss North America,Eden Prairie, MN; AOAC, 1990) [18]. Please refer to Line 150-152, Line 162-165.
Comment: Lines 154-160: Statistical analyses were performed and described correctly. However, the authors should add and describe the statistical models used.
Response: Revised. Please refer to Line 183-184. We have added and described the statistical models used, which was “The data were performed using the PROC MIXED of SAS with sows or newborn piglets as the experimental unit.”
Comment: Results
Lines 163-165: The description in Tables 2 and 3 is incomplete and needs to be improved. The authors should also describe the variables for which no statistically significant effects were detected.
Response: Revised. Please refer to Line 195-196. We have complete the description about all the Tables, we mentioned the no significant difference between two group of the parameters.
Comment: Line 175: The abbreviation ATTD should be changed to the full name. This correction applies to all titles and subtitles.
Response: Revised. Please refer to Line 163-164 and Line 211-212. We have changed the abbreviation ATTD into apparent total tract digestibility.
Comment: Lines 176-225: The interpretation and description of the results tables is very well done. Just a minor correction in this section, the authors should also describe the variables for which no statistically significant effects were detected.
Response: Revised. Please refer to Line 226-229, Line 240-243. We have added the described the variables for which no statistically significant effects were detected.
Comment: Discussion
Lines 226-323: The entire discussion section is very well-written and organized. Furthermore, the discussion contains the necessary depth to explain in detail the results obtained (although little information is available on the specific topic evaluated). Therefore, I have no additional corrections in this section.
Response: Revised. Thanks for the comments. We have improved it to make it better.
Comment: Conclusions
Lines 325-327: The conclusion drawn by the authors is consistent with the title, objectives, and results of the manuscript. However, abbreviations should not be used in this section. Please correct.
Response: Thanks for the suggestion. We have added the full name of abbreviations. Please refer to Line 390-395. We changed the conclusion into “In conclusion, dietary 25-hydroxycholecalciferol supplementation from d 85 of gestation increased protein and fat contents in colostrum and average birth body weight of newborn piglets, enhanced apparent total tract digestibility of crude protein in sows. Besides, diet supplemented with 25-hydroxycholecalciferol also improved antioxidant capacity, immunoglobulins, and calcium and phosphorus related biochemical properties in sows and newborn piglets.
Reviewer 3 Report
Comments and Suggestions for Authors
The methods part needs to be reorganized, for example, the authors need to insert the measurements of the blood after the collection of blood samples, the same for the colostrum, digestibility trail,….
Line 85, how did you adjust the day of the gestation to be 85 days?
Line 90, please remove “or exceeded”.
Title of table 1, please replace “nutrient” with “chemical”.
Table, 1 please calculate all the values in this table to be as dry matter basis, they cannot be as fed basis. Also, please add the values of OM, NDF, EE, chromium,…
Line 91, is this diet was used through all the experimental period (gestation and after farrowing)? If so please clarify.
Line 105, fed what?
Line 106, again fed what?
Line 115 are these 4 reproductive glands from all animals?
Line 118, why you collected the fecal samples on day 107? the collection number of the fecal samples are not clear, is it was collected once at say 107 or six times as you wrote in line 125, or it refer to the number of animals, if so, may one day is not sufficient to measure the digestibility in all over the gestation period.
Line 121, pooled for what? for animal or treatment?
How did you measure the lactose, CP, and fat of the colostrum in the part of methods?
Please insert the full name of ATTD, and check the full names of all abbreviations at their first mentions.
The conclusion is weak, It needs to be improved.
Author Response
Dear Sir
Good day. Thank you very much for your kind consideration with our submitted article and offering us the further opportunity to submit the revised manuscript. Please find here the point to point expert reviewer’s and editor’s comments with necessary changes as per suggested with this attached file. We have revised our manuscript for language and grammar checked by a native English speaker working in our University, and highlight the change with red color. We do thanks to skilled reviewers, academic editors and editorial board members as well for their critical evaluation to make the manuscript more effective for review process in Animals Journal.
Many thanks.
Sincerely yours,
Prof. Dr. Xiang Shu Piao,
Beijing Jingwa Agricultural Science and Technology Innovation Center
College of Animal Science and Technology, China Agricultural University, Beijing 100193, China
Corresponding Author,
Email: piaoxsh@cau.edu.cn.
Reviewer 3:
Comment: The methods part needs to be reorganized, for example, the authors need to insert the measurements of the blood after the collection of blood samples, the same for the colostrum, digestibility trail,….
Response: Thanks for the comments, we have inserted the measurements of the blood after the collection of blood samples, the same for the colostrum, digestibility trail. Please refer to 150-166. For example, we added “The contents of protein, lactose, and fat in colostrum samples were measured via a Milkoscan System 4000 (Foss North America,Eden Prairie, MN; AOAC, 1990) [18].” and “Neutral detergent fiber (NDF) was determined using fiber analyzer (Ankom Technology, Macedon, NY, USA). The calculation used for apparent total tract digestibility (ATTD) of nutrient was conducted following Li et al (2019) [20].”
Comment: Line 85, how did you adjust the day of the gestation to be 85 days?
Response: Thanks for the question, we adjusted the day of the gestation based on farrowing day, normally d 114 of gestation.
Comment: Line 90, please remove “or exceeded”.
Response: Revised. Please refer to Line 110.
Comment: Title of table 1, please replace “nutrient” with “chemical”.
Table, 1 please calculate all the values in this table to be as dry matter basis, they cannot be as fed basis. Also, please add the values of OM, NDF, EE, chromium,…
Response: Thanks for the suggestion, we have mostly revised according to the advice. Please refer to Table 1. Normally we made formula as-fed base. We added the analyzed values of OM, NDF, EE because they are nutrients.
Comment: Line 91, is this diet was used through all the experimental period (gestation and after farrowing)? If so please clarify.
Response: We have added this statement. Please refer to Line 110, This diet was used through all the experimental period.
Comment: Line 105, fed what?
Response: Fed diets, please refer to Line 124-125. Before farrowing, sows were fed diets at 08:00 a.m. and 16:00 p.m.
Comment: Line 106, again fed what?
Response: Fed diets, please refer to Line 125-126. After farrowing, sows were fed diets ad libitum.
Comment: Line 115 are these 4 reproductive glands from all animals?
Response: The 4 reproductive glands are from one animals.
Comment: Line 118, why you collected the fecal samples on day 107? the collection number of the fecal samples are not clear, is it was collected once at say 107 or six times as you wrote in line 125, or it refer to the number of animals, if so, may one day is not sufficient to measure the digestibility in all over the gestation period.
Response: Thanks for the suggestion, we have revised according to the advice. After about 25-28 days, which was around d 107-109, we collected fecal sample for 3 days to measure ATTD of nutrients. Please refer to Line 138-142. From d 107-109 of gestation, the garb sampling technique and rectal palpation were used to collect fresh fecal samples of sows (about 2 kg feces/sow; n = 6) closed to the average parity and BW from 3 consecutive days. The feces of individual sow of 3 days were collected individually and dried at 65 °C for 72 h.
Comment: Line 121, pooled for what? for animal or treatment?
How did you measure the lactose, CP, and fat of the colostrum in the part of methods?
Please insert the full name of ATTD, and check the full names of all abbreviations at their first mentions.
Response: Revised. Pooled or collected for animals. Please refer to Line 142-143. The feces of individual sow of 3 days were collected individually and dried at 65 °C for 72 h. Each sow fecal sample was collected and dried. The contents of protein, lactose, and fat in colostrum samples were measured via a Milkoscan System 4000 (Foss North America,Eden Prairie, MN; AOAC, 1990) [18]. We also insert the full name of ATTD and check the full names of all abbreviations at their first mentions.
Comment: The conclusion is weak, It needs to be improved.
Response: Revised. Please refer to Line 390-395. In conclusion, dietary 25-hydroxycholecalciferol supplementation from d 85 of gestation increased protein and fat contents in colostrum and average birth body weight of newborn piglets, enhanced apparent total tract digestibility of crude protein in sows. Besides, diet supplemented with 25-hydroxycholecalciferol also improved antioxidant capacity, immunoglobulins, and calcium and phosphorus related biochemical properties in sows and newborn piglets.
Round 2
Reviewer 3 Report
Comments and Suggestions for Authors
Congratulation for the work, the manuscript has been improved, only few comments have to be addressed first before the final recommendation:
Line 59, please replace “in feed for” with “in diets of”
Line 81, this sentence needs English editing.
Line 100, is this study was published, if so please insert the reference.
In table 1, please add the DM, OM, EE and NDF after the CP.